# Pathologic Changes in and Immunophenotyping of Polymyositis in the Dutch Kooiker Dog

**DOI:** 10.3390/ani14172519

**Published:** 2024-08-29

**Authors:** Vanessa Alf, Yvet Opmeer, G. Diane Shelton, Guy C. M. Grinwis, Kaspar Matiasek, Marco Rosati, Paul J. J. Mandigers

**Affiliations:** 1Section of Clinical and Comparative Neuropathology, Centre for Clinical Veterinary Medicine, Ludwig-Maximilians-Universität Munich, 80539 Munich, Germany; v.alf@dekanat.vetmed.uni-muenchen.de (V.A.); kaspar.matiasek@neuropathologie.de (K.M.); marco.rosati@outlook.de (M.R.); 2Expertise Centre of Genetics, Department of Clinical Sciences, Faculty of Veterinary Medicine, Utrecht University, 3584 CM Utrecht, The Netherlands; y.opmeer@uu.nl; 3Department of Pathology, School of Medicine, University of California, San Diego, CA 92093-0709, USA; gshelton@health.ucsd.edu; 4Department of Biomolecular Health Sciences, Faculty of Veterinary Medicine, Utrecht University, 3584 CL Utrecht, The Netherlands; g.c.m.grinwis@uu.nl

**Keywords:** breed, immune-mediated disease, IBA-1, inflammatory myopathy, hereditary disease

## Abstract

**Simple Summary:**

In 2023, we described the clinical signs and histology of an inflammatory myopathy that affects Kooiker dogs, one of the nine Dutch breeds. In this additional study, we further investigated muscle changes in the affected dogs. Our results showed typical signs of inflammation, cell infiltration into muscle tissue, and immunophenotyping that revealed the involvement of both adaptive and innate immune responses, with T-cells being the predominant cell type. Additionally, histological analysis highlighted muscle damage and regeneration. Based on our results, we conclude that this is polymyositis (PM). PM in these dogs shares similarities with other breeds but appears to have unique characteristics. The findings described contribute to understanding PM in Kooiker dogs and its immune-mediated mechanisms in general.

**Abstract:**

Earlier, we described a breed-specific inflammatory myopathy in Dutch Kooiker dogs (Het Nederlandse Kooikerhondje), one of the nine Dutch breeds. The disease commonly manifests itself with clinical signs of difficulty walking, muscle weakness, exercise intolerance, and/or dysphagia. In nearly all dogs’ creatine kinase (CK) activity was elevated. Histopathology reveals the infiltration of inflammatory cells within the skeletal muscles. The objective of this study was to further investigate and characterize the histopathological changes in muscle tissue and immunophenotype the inflammatory infiltrates. FFPE fixed-muscle biopsies from 39 purebred Kooiker dogs were included and evaluated histopathologically according to a tailored classification scheme for skeletal muscle inflammation. As in other breed-related inflammatory myopathies, multifocal, mixed, and predominantly mononuclear cell infiltration was present, with an initial invasion of viable muscle fibres and the surrounding stroma leading to inflammation, necrosis, and tissue damage. Immunophenotyping primarily revealed lymphohistiocytic infiltrates, with CD3+ T-cells being the predominant inflammatory cell type, accompanied by CD8+ cytotoxic T-cells. The concurrent expression of MHC-II class molecules on myofibres suggests their involvement in initiating and maintaining inflammation. Additionally, CD20+ B-cells were identified, though in lower numbers compared to T-cells, and IBA-1-positive macrophages were frequently seen. These findings suggest a breed-specific subtype of polymyositis in Kooiker dogs, akin to other breeds. This study sheds light on the immune response activation, combining adaptive and innate mechanisms, contributing to our understanding of polymyositis in this breed.

## 1. Introduction

‘Het Nederlandse Kooikerhondje’, or Dutch Kooiker dog, one of the nine Dutch breeds, is a FCI (Fédération Cynologique Internationale, registration number 314)-recognized duck decoy dog breed from the Netherlands. After the Second World War, this breed was almost extinct and was re-established by carefully selecting a small number of Kooiker dogs [1,2]. The current Kooiker dog population descends from approximately 10 dogs [3,4,5].

Due to this small population size and the close relationship of the individual (breeding) dogs, the likelihood of inbreeding and the associated hereditary diseases are obvious. The homozygosity of the parents and thus of the litter, as well as strong and strict selection programs, resulted in high inbreeding levels and increased the risk of genetic disorders [3,6,7]. Up to now, two genetic disorders have been described in this breed: von Willebrand’s disease [8,9] and hereditary necrotizing myelopathy [2,10]. Several other possibly hereditary diseases have been identified, including cataract [5], patellar luxation [11], and epilepsy [12,13]. Recently, we added to this list an inflammatory myopathy [4].

Canine polymyositis (PM) is an immune-mediated neuromuscular disorder characterized by multifocal, polyphasic infiltration of the mononuclear inflammatory cells, predominantly CD8+ T-cells, into striated muscles [14,15,16,17]. It is a subgroup of immune-mediated inflammatory myopathies (IMs) and, together with dermatomyositis (DM), belongs to the generalized forms. Masticatory muscle myositis (MMM) and extraocular myositis (EOM) represent focal expressions of IMs [17,18,19]. PM has been identified in Hungarian Vizslas, Boxer dogs, German Shepherd dogs, Newfoundlands, and Labrador and Golden Retrievers. In Boxer dogs and Newfoundlands sarcolemma-specific autoantibodies have been detected, and in Vizslas, a certain MHC-II haplotype is associated with the disease [17,18,20,21,22,23,24]. Clinicopathologic investigation may reveal increased muscle enzymes, and electromyography can confirm localization to the skeletal muscles. Nevertheless, the gold standard for diagnosis is a muscle biopsy with the identification of lymphocytic infiltrates directed towards muscle fibres and otherwise intact muscle fibres [17,18]. Histopathological findings in PM consist of a multifocal, mixed predominantly mononuclear cell infiltration of varying severity depending on the sampled muscles and stage of the disease. Degenerative features, including myofibre diameter variations with fibre atrophy, nuclear internalization, necrosis, myofibre loss and fibrosis, and regenerative changes like nuclear rowing, compensatory hypertrophy and regenerating fibres, are frequently seen in the muscles of affected animals [16,18,19,23,25]. The composition of inflammatory infiltrates in PM includes lymphocytes, predominantly of the T-cell type with cytotoxic CD8+ T-cells, in greater number than CD4+ T-cells, and lesser numbers of dendritic cells, macrophages, and B-cells. The CD8+ lymphocytes are distributed along the endomysium surrounding necrotic myocytes and invading viable muscle fibres. In contrast, in MMM, CD4+ T-cells and B-cells are more commonly reported than CD8+ T-cells, and their distribution is concentrated along the perimysium and perivascularly [16,17]. Increased expression of MHC-I on the muscle sarcolemma of infiltrated and non-infiltrated myofibres [16,26], as well as the expression of MHC-II on infiltrating cells and on the sarcolemma and sarcoplasm of the infiltrated and non-infiltrated myofibres, is frequently observed [16,25,27,28,29].

The aim of this study is to describe and characterize the pathological changes in and the immunophenotype of the inflammatory infiltrates in this breed-related inflammatory myopathy in the Dutch Kooiker dog.

## 2. Materials and Methods

### 2.1. Study Population and Tissue Collection

A total of 39 purebred Kooiker dogs, a subset of the 160 Kooiker dogs described earlier [4], were available for this study. Skeletal muscle samples from thirty-five dogs were submitted by the last author (PJJM, a diplomate of the European College of Veterinary Neurology (ECVN)), and muscle samples from four Kooiker dogs were sent directly to the Neuropathology laboratory, Ludwig-Maximilians-Universität Munich, Germany (http://neuropathologie.de/index.html, accessed on 1 July 2022), for histopathological examination. All owners agreed to participate in this study and gave informed consent. All Kooiker dogs included had been diagnosed with the inflammatory myopathy specific for this breed, as described earlier [4]. The selected dogs all had clinical signs of locomotion problems, dysphagia, or a combination of these signs. The CK activity was, in all dogs, above the reference range of 200 U/L, and histopathology confirmed, in all dogs, inflammatory myopathies. Muscle biopsies were obtained as described earlier [4]. All biopsies were from mostly the appendicular muscles m. triceps brachii (TRI) and m. biceps femoris (BF), but a small number were taken from the masticatory and axial muscles as well. All samples were processed as described earlier [4].

### 2.2. Histopathological Evaluation

All FFPE (formalin-fixed paraffin-embedded)-slides from collected tissues were investigated by two diagnosticians (M.R., Dipl. ACVP, K.M.; AM-ECVN, V.A. doctoral student) by light microscopic examination (Zeiss Axiophot^®^, Zeiss Instruments, Oberkochen, Germany) according to standard diagnostic algorithms for the interpretation of neuromuscular biopsies. Whenever possible, sections of skeletal muscles were interpreted in transverse and longitudinal orientation of muscle fibres. Secondly, myopathological changes in and the amount and type of inflammatory infiltrates were recorded. A tailored grading scheme for skeletal muscle inflammation on FFPE- slides stained with H&E (hematoxylin-eosin) and Giemsa including 91 histological variables was applied. In brief, the scheme focused on the lesion type, topographic distribution, cell type, and degree of severity of inflammatory infiltrates. Semiquantitative scores were applied for each variable by a board-certified pathologist and a trainee in pathology (M.R., Dipl. ACVP; V.A. doctoral student) from the Section of Clinical and Comparative Neuropathology, LMU, Munich. A semiquantitative assessment of the (immune-) staining of inflammatory infiltrates was performed considering their quantity and distribution.

Descriptions of each histological variable and its respective score or grading are summarized in Appendix A.

### 2.3. Immunohistochemistry

Immunohistochemistry was performed on all muscles sampled for immunophenotyping inflammatory infiltrates with the following antibodies: rabbit anti IBA-1 (concentration: 1:1000, Abcam ab 108530: https://www.abcam.com/en-nl), rabbit anti CD3 (concentration: 1:100, Dako A0452), rabbit anti CD8 (concentration 1:500, Linaris PAK0129), and rabbit anti CD20 (concentration: 1:1000, Thermo Scientific RB-9013). Immunohistochemical staining of MHC-II in muscle fibres was performed with a mouse anti MHC-II (concentration: 1:200, Dako M0746).

Sections were taken from paraffin blocks with thicknesses of 3–5 µm. After deparaffinization and rehydration through xylol and a graduate alcohol series, the slides were rinsed with distilled water and, except for CD20, boiled for 20 min in the microwave for antigen retrieval. For CD3, CD8, and IBA-1, a citrate buffer (pH = 6) was used; for MHC-II, an EDTA-buffer (pH = 9) as used. To quench endogenous peroxidase activity, the tissue was treated with H_2_O_2_ for 30 min at room temperature. After rinsing in PBS, normal horse serum 2.5% (ImmPRESS^®^-HRP Horse Anti-Rabbit/Anti-Mouse IgG Polymer Detection Kit, MP-7401 and MP-7402, Vector Laboratories: https://vectorlabs.com/, accessed on 26 August 2024) was used for blocking. The sections were incubated with the primary antibody at 4 °C overnight. Next, they were washed with PBS and incubated with the secondary antibody (ImmPRESS^®^-AP Horse Anti-Rabbit/Anti-Mouse IgG Polymer Detection Kit, MP-7401 and MP-7402, Vector Laboratories) for 30 min at room temperature. The sections were rinsed again, and colour reaction was localized with 3,3-di-amino-benzidine tetrahydrochloride (ImpactDAB^®^ Substrate Peroxidase, SK-4105, Vector Laboratories). Then, they were counterstained with haematoxylin, dehydrated with an alcohol series, cleared in xylol, and covered. For quality assessment, positive and negative controls for each primary antibody on canine lymph node tissue were run in parallel. Similarly to the other histological variables Appendix A, the topographic distribution of each cell type and their severity were recorded through semiquantitative scores, and the same was applied for the expression of MHC-II in non-infiltrated and viable muscle fibres.

### 2.4. Detection of CD4-Positive T-Cells by ISH

In situ hybridization (ISH) using RNAscope^®^ was performed for a subset of 10 randomly selected cases of these Kooiker dog samples to evaluate the presence and distribution of CD4-positive T-cells. An RNA probe targeting the mRNA in canine CD4+ T-cells was used. As a positive control, a probe targeting the mRNA of the peptidyl-prolyl-isomerase B housekeeping gene, which is commonly observed and widely expressed, was utilized (RNAscope™ Probe-Pci-PPIB; cat no. 868,241), while the negative control consisted of a probe targeting a bacterial dihydropicolinate reductase (RNAscope™ Negative Control Probe-DapB; cat no. 310,043). Thymus tissue of an otherwise healthy dog served as positive (antigen) and negative (reagent) controls. CD4-expressing cells were visualized using ISH with the RNAScope™ Probe-CI-CD4 (cat no. 459,551) according to the manufacturer’s instructions. In brief, FFPE muscle sections were deparaffinized and subjected to pre-treatment with 1× Target Retrieval solution and RNAscope^®^ Protease Plus solution. Following this, the tissue underwent a sequence of pre-amplification and amplification steps, succeeded by the administration of a chromogenic substrate. The specimens were finally counterstained using Hematoxylin Gill No. 2 (Merck, Darmstadt, Germany).

The evaluation and scoring of the signals was performed as described in Appendix A.

### 2.5. Statistical Analysis

The results were analysed with R Statistical software (version 4.2.3). Descriptive statistics (number of dogs, mean ± standard deviation, median) and chi-squared tests were performed. The results were considered significant when the *p*-value was <0.05.

## 3. Results

### 3.1. Study Cohort

Our Kooiker cohort included 12 intact females, 7 spayed females, 15 intact males, and 5 neutered males. Information about the age at onset of clinical signs related to the later diagnosed myopathy were available for 39 dogs and ranged from 1 to 10 years (mean: 4.29 ± 2.35 years; median: 4.0 years). In 30 dogs, the age at death was known, so the duration of disease could be calculated. The time sick ranged from 0.1 to 4.9 years (mean: 1.5 ± 1.65 years; median: 0.9 years). The actual cause of death was not known, but information on the outcomes of the other nine dogs is available: eight of them died and one is still alive. Pathologic materials were collected over a period of 23 years.

In 33 cases, the exact biopsy sites were stated, and they are summarized in Table 1.

### 3.2. Histopathological Findings in HE and Giemsa

All cases showed inflammatory changes in one or more of the sampled skeletal muscles. The overall grade of myositis was considered mild in 8 (21%), moderate in 17 (44%) and severe in 14 (36%) dogs. The distribution of inflammatory cells was predominantly myofiber-directed and interstitial in all cases. In 24 (62%) dogs, the coring-out of fibres (Figure 1) could be identified, and in 4 (10%) cases, infiltrative cells were also detected in the fascia or with a diffuse pattern across muscle fascicles. Leukostasis was seen in 15 (38%) cases, but vasculitis was not observed in this cohort.

All cases displayed several myopathic changes. Variation in myofibre diameters was present in all dogs graded as mild in 19 (49%), moderate in 18 (46%), and severe in 2 (5.1%) cases. A polygonal atrophic pattern was present in all cases, classified as mild in 11 (28%), moderate in 23 (59%), and severe in 5 (13%). Angular atrophy was much rarer, and only seven (18%) dogs were mildly affected, whereas the rest were inconspicuous. Hypertrophy of single muscle cells was observed in 10 cases, with 7 (18%) mildly and 3 (7.7%) moderately affected. Myofibre splitting was seen in two (5%) dogs and mineralization in one (2.6%) dog, and the nuclear position within muscle cells was considered normal in all cases.

All Kooiker dogs had polyphasic myonecrosis, and muscle fibre regeneration was seen in 21 (54%) of them. The related stroma was normal in 20 (51%) cases and displayed mild fibrosis in 11 (28%) and moderate fibrosis in 8 (21%) cases. Lipomatosis was seen in 11 (28%) as mild, in 9 (23%) as moderate, and in 2 (5%) as severe.

Intramuscular nerve branches were seen in 26 (67%) dogs and were mostly considered normal. One case showed mild endoneurial lymphocytic inflammation. Endoneurial and perineurial vasculitis were not observed in any cases.

Plasma cells, neutrophils, eosinophils, and mast cells were evaluated on HE- and Giemsa-stained slides. Neutrophils were found in 25 (64%) of the samples and represented 1–25% of the total inflammatory cell population (grade 1). Their distribution was fibre-directed in all cases, with the involvement of the endo- and perimysium in 13 (52%) cases. Only one case displayed perivascular interstitial neutrophilic infiltration.

Eosinophilic granulocytes were represented in low numbers in 11 (28%) cases, being mainly fibre-directed in 10 (91%) and interstitial in 7 (64%) subjects. Low and moderate amounts of plasma cells were observed in 28 dogs (72%) and 1 dog (2.6%), respectively, and these were mainly fibre-directed in 27 (93%) and interstitial in 12 (41%) dogs. Nine (31%) cases also displayed perivascular infiltration, and three (10%) cases displayed plasma cells in the fascia. Nearly all dogs had mast cell infiltrates, being of a low degree in 32 (82%) and moderate in 5 (13%) cases. All mast cells were interstitial, with 31 (84%) being perivascular and 22 (59%) being within the fascia, and 1 (2.6%) case each displayed a random or fibre-adjacent distribution.

A comparison of the distribution of the different inflammatory infiltrates is visualized in Figure 2.

### 3.3. Histopathological Findings in IHC and ISH

The T-cell marker CD3 (Figure 3) was expressed in 38 (97%) of cases, mainly scored as grade 2 in 22 (56%) cases, followed by grade 3 in 10 (26%) case and grade 1 in 6 (15%) cases. In all of them, the positive cells were mainly fibre-directed, with interstitial involvement in (37 cases; 97%). Only 4 (11%) cases presented CD3+ T-cells within the fascia and in 1 (3%) case in the perivascular space.

CD8-positive cytotoxic T-cells were identified in 24 (61%) dogs and scored as grade 1 in 21 (88%) and grade 2 in 3 (12%) cases, and their distribution within the tissue was always fibre-directed and interstitial in 22/24 (97%) of the positive cases. The fascia and the perivascular space were not infiltrated (Figure 3 and Figure 4).

CD20+ B-cells were detected in all dogs (Figure 5), with 35 (90%) being grade 1 and 4 (10%) being grade 2. All of them had interstitial infiltrates, and in 14 (38%) cases, they also appeared fibre-directed. In three (8%) cases, the CD20+ cells were also found around blood vessels.

Immunohistochemistry for MHC-II (Figure 6 and Figure 7) revealed a positive signal of sarcolemmal MHC-II staining in 34/39 cases distributed among the following grades/scores: 15 (38%) were grade 1, 15 (38%) were grade 2, and 4 (10%) were grade 3.

Macrophages were identified using the IBA-1 marker, and nearly all dogs had IBA-1-positive infiltrating cells, mainly scored as grade 2 in 34 (87%) cases, followed by grade 3 in 3 cases (7%) and grade 1 in 1 patient (3%). IBA1+ cells were fibre-directed in 37 (97%), interstitial in 37 (97%) and within the fascia in 1 (3%) of the dogs.

For all findings in detail, see Appendix A.

A subset of 10 cases was evaluated using ISH to visualize CD4-positive T-cells. A specific ISH signal was seen in all cases with varying intensity and distribution. In 9/10 cases, the overall density was considered low, and in the remaining case, it was moderate. All CD4-positive cells were interstitial, had a fibre-invasive pattern, or were samples in which the involvement of the fascia or the perivascular space was not observed.

## 4. Discussion

In this study, we characterized the pathological changes in axial and perpendicular muscle biopsies in 39 Kooiker dogs affected by an inflammatory myopathy, consistent with polymyositis. The hallmarks of this condition are multifocal, mixed, predominantly mononuclear cell infiltrates directed to viable myofibres and surrounding stroma (fascia, endomysium and perimysium) affecting multiple skeletal muscles, similarly to what is described in other canine IMs [15,16,17,19]. The coring-out of fibres, as a sign of early invasion of otherwise intact myofibres, was seen in most of the cases and, together with polyphasic myonecrosis, suggests that myofibres are the main target of the immune-mediated reaction in this breed. Immunophenotyping demonstrated that the infiltrates found in our Kooiker population were predominantely lymphohistiocytic, with CD3+ T-cells being the most frequent inflammatory cell type, alongside CD8+ cytotoxic T-cells and the expression of MHC class II molecules on myofibres. Additionally, CD20+ B-cells and CD4+ T-cells were detected, albeit in lower numbers compared to CD3+ and CD8+ T-cells, and IBA-1-positive macrophages were also frequently observed. Sarcolemma-specific autoantibodies have been already identified in other canine breeds affected by PM, but they have not been investigated so far in Kooiker dogs [15,21,24].

Immunophenotyping through histochemical and immunohistochemical stains reveals a combination of adaptive and innate immune response activation. In humans and dogs, there is strong evidence for antigen processing and presentation within skeletal muscles leading to an aberrant immune response exploiting both cell-mediated and humoral effector mechanisms [16,18,30,31]. Research work has mainly focused on the role of the adaptive immune system in the involvement of lymphocytes in the pathogenesis and progression of IMs, but the contribution of the innate immune system is less well understood [32,33].

The predominant inflammatory cell population in PM in Kooiker dogs is represented by lymphohistiocytic infiltrate, with CD3+ T-cells being the most frequently encountered cell type, similar to what was reported in other breeds [16,18,34,35]. T-lymphocytes are a crucial part of the adaptive immune system of vertebrates, as they recognize antigenic peptides presented by MHC class molecules and thereby drive and regulate the cell-mediated immune response [36,37]. All forms of immune-mediated myopathies in humans and animals have the cell-mediated myocytotoxic activity of T-cells in common [18]. Through immunohistochemistry on FFPE sections, we identified CD8+ cytotoxic T-cells in two thirds of our patients, making them one of the principal causes of damage to the muscle fibres. Antigens presented on MHC-I class molecules on the surface of the target cell activate CD8+ T-cells and initiate the release of cytotoxic molecules leading to cell death, tissue destruction and epitope spreading of autoantigens [28,33]. Neither the initial trigger of the CD8+ T-cell infiltration into muscles nor the cause of the sarcolemmal upregulation of MHC-I is known [33]. Previous reports of canine PM also described a predominance of CD8+ T-cells over CD4+ helper T-cells, while in the course of MMM, CD4+ cells were the predominant T-cell population in close proximity to B-cells [16,17]. CD4+ T-cells play a crucial role in initiating and shaping adaptive immune responses by recognizing peptides presented on antigen-presenting cells (APCs) via MHC-II molecules. They produce cytokines activating other immune cells such as natural killer cells and macrophages, upregulate MHC expression, and induce immunoglobulin isotype switching in B-cells [38,39,40]. In our subset of 10 cases created for the evaluation of CD4+ T-cells using ISH, the grade was considered mild, with interstitial distribution and the sparing of the fascia or the perivascular area. As previously described by Pumarola et al. [16], they were smaller in number than CD8+ cells. In this regard, further immunohistochemical evaluation of CD4 and MHC-I expression could not be pursued because of the lack of fresh frozen samples and limited affinity of commercially available primary antibodies for FFPE canine tissues.

The predominance of cell-mediated immunity is also reflected in the expression of MHC-II class molecules that are normally expressed in APCs. In healthy skeletal muscles, MHC-II is not expressed on the sarcolemma, and its immunohistochemical detection is limited to the endothelial cells of blood vessels [41]. All nucleated cells are able to express MHC-I, while MHC-II expression is normally restricted to professional APCs and can be detected on other cell types following inflammatory signals [42]. The increased expression of MHC-I and expression of MHC-II in myofibres of IM-affected dogs has been described in cases of MMM, PM, and infectious myositis caused by *Neospora caninum* [16,27,28,35]. MHC-I expression is a diagnostic tool for identifying IMs with high sensitivity but low specificity [29,43]. MHC-II staining, in contrast, can distinguish IMs from hereditary myopathies in humans, while MHC-I is expressed in both disease groups and in all myopathies with autophagy [29]. This is strongly supportive of the active role of MHC-II in initiating and maintaining inflammation, rather than just being a byproduct of myositis [44].

CD20+ B-cells were identified in 90% of the samples, though in lower amounts compared to T-cells, different to what was previously published on B-cell infiltration in canine PM [16,35]. B-cells are responsible for antigen-specific humoral immune response, and their role in IMs remains unclear. Skeletal muscle antigen-specific autoimmunity, the maintenance of serum antibody-levels, and the persistence of autoantibody production are suggested in human myositis [45]. The local and rapid maturation of B-cells to antibody-producing plasma cells is described in inflamed muscles, and there is evidence that the inflammatory environment in IMs supports this maturation [45,46]. In this cohort, low percentages of plasma cells were also recorded in 75% of patients, similar to the findings reported in human PM, where a local antigen-driven response was demonstrated [47]. Genes participating in B-cell maturation, growth, migration, and activation are significantly upregulated in MMM, as well as in PM [35], adding further complexity into the underlying mechanisms at the base of autoimmunity. In contrast to PM, B-cells are regularly found in MMM, as well as autoantibodies against type 2M muscle fibres, which are unique to the masticatory muscles innervated by the trigeminal nerve, including those for jaw closure and planar mandible movements (temporalis muscle, masseter muscle, pterygoid muscle) and those for jaw opening (rostral digastricus muscle) [16,48].

IBA-1 is a protein expressed in macrophages and microglia that is upregulated during the inflammation and activation of these cells [49,50]. Young macrophages infiltrate the muscle tissue early in the disease for phagocytosis and antigen presentation, while mature macrophages, in later stages, appear crucial for muscle regeneration by ensuring the clearance of cellular debris, and by supporting satellite cell proliferation and differentiation [49,51,52]. In our cohort, IBA-1+ macrophages were consistently detected and represented the second most common inflammatory cell type after lymphocytes. Further characterization of histiocytic subtypes was out of the scope of this investigation, but it would be crucial for understanding their contribution to the pathogenesis of IMs. Macrophages can polarize into distinct phenotypes based on their activation status, and the two main polarization states are proinflammatory (M1) and anti-inflammatory (M2) macrophages [53,54]. The balance between M1 and M2 responses is crucial for effective immune responses and tissue repair [49,55,56]. The disturbance of this balance can contribute to various diseases, including chronic inflammatory conditions, autoimmune diseases, and tissue fibrosis [53,57].

The relevance of innate immunity in the context of PM in Kooiker dogs appears secondary and is most likely due to general response to tissue injury. Damage-associated molecular patterns (DAMPs), also known as alarmins, are molecules released by stressed cells undergoing necrosis and acting as endogenous danger signals to promote and exacerbate the inflammatory response [58,59]. In IMs, the overexpression of proinflammatory cytokines mediates the chemotaxis of neutrophils to the affected area [60,61]. Likewise, 64% of patients in this cohort showed neutrophilic infiltration, though they were graded as very mild (mainly grade 1). Eosinophils are also observed during the course of PM (28% in the present investigation) and may contribute to muscle fibrosis at chronic stages [32], and they may also play a role in tissue damage due to their cytotoxic action [62]. Low amounts of mast cells were frequently encountered in up to 82% of Kooiker dogs. In a study investigating the role of mast cells, the total number of mast cells in human skeletal muscle was increased in PM but not in DM [63]. While its primary role is to defend against pathogens and promote healing, mast cell activation can also contribute to the development and exacerbation of autoimmune diseases through several mechanisms [64].

Besides inflammation, histology highlighted a series of myopathic changes most likely secondary to inflammation and/or reduced muscle fitness and overall physical activity. All cases featured myofibre diameter variations with round polygonal atrophic and/or less commonly angular atrophic fibres. Atrophic muscle fibres forfeit mass and contractile function. In human and canine IMs, polygonal and angular atrophy of myofibres is frequently seen, as are hypertrophic fibres compensating for the reduction in the contractile activity of adjacent atrophic fibres [65,66,67,68]. About half of our cases were affected by the loss of functional skeletal muscle due to replacement by fibrosis and lipomatosis, which occurred either individually or side by side, indicating a long-lasting inflammatory condition. Nevertheless, regenerating myofibres were observed in 54% of cases.

PM has been described in other canine breeds, mainly larger breeds such as Boxers, Labrador Retrievers, Golden Retrievers, Dobermanns, German Shepherd dogs, Newfoundlands, and Hungarian Vizslas [23]. The affected muscles and the clinical signs vary among the different breeds. In Labrador Retrievers, Golden Retrievers, and Newfoundlands, the appendicular and axial muscles are predominantly involved, whereas in Hungarian Viszlas, it is mainly the masticatory, pharyngeal, and oesophageal muscles that are affected [17,18,20]. In Boxers, PM appears to involve striated muscles in a more generalized manner, and a link to malignancy is suspected to be a possible manifestation of pre- and paraneoplastic syndromes. Despite this correlation with neoplasia, histopathological changes in Boxer PM did not differ from the IMs of other breeds [17,21,69].

In Kooiker dogs, there is no predilection for specific skeletal muscle groups. Locomotory problems are the main clinical sign, next to dysphagia. In the earlier described cohort of 160 Kooiker dogs [3], biopsies of masticatory muscles from 6 patients revealed no inflammatory changes, and an overlapping MMM seems rather unlikely [4]. Based on the clinical signs and the distribution of the lesions, PM in Kooiker dogs is comparable to PM in breeds like Labrador Retrievers, Golden Retrievers and Newfoundlands [4,18]. Whether these breeds share a common pathogenesis and/or common antigenic targets remains to be further elucidated. A study on the MHC-II haplotype in Hungarian Vizslas provides evidence of a genetic basis for the development of PM due to the possible higher susceptibility to aberrant responses of the immune system against self-antigens. The penetrance of the identified risk haplotype was, however low; hence, the authors concluded that other genetic and environmental factors may also be involved [22]. Previous reports identified several genes associated with immune cell function, the regulation of MHC-II expression, and increased pro- and anti-inflammatory cytokine and chemokine expression associated with MMM and PM [35]. More recently, another Dutch dog breed, the Dutch Shepherd Dog, has drawn attention because of a possible genetic form of PM. In a very small number of closely related dogs, a mutation within the mitochondrial aspartate/glutamate carrier was identified [70]. This mutation has been deemed accountable for creating a muscle environment that promotes inflammation, ultimately resulting in the onset of PM [70].

## 5. Conclusions

In conclusion, we characterized the histopathologic features of PM in Kooiker dogs, providing some useful insight into the immunophenotyping of its inflammatory infiltrates and associated myopathic changes. These findings may provide the basis for a better understanding of the dysregulation of the immune system and help us in understanding the molecular and genetic background of this disease.

## Figures and Tables

**Figure 1 animals-14-02519-f001:**
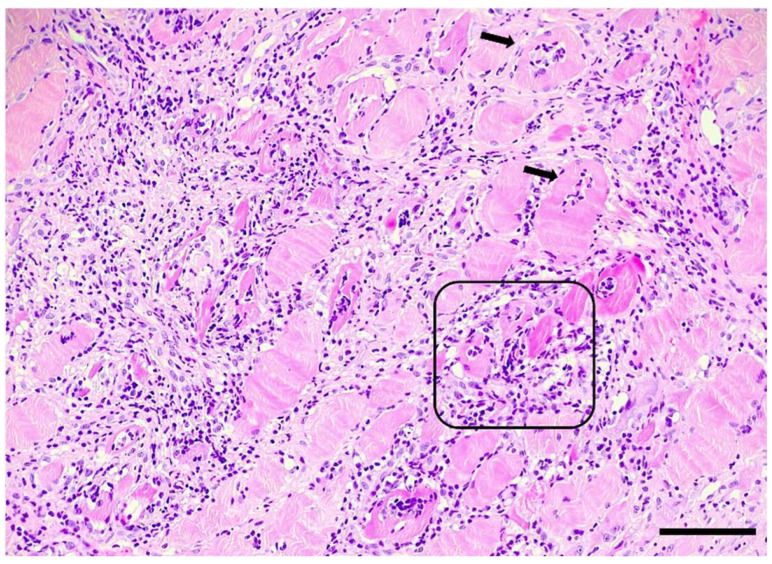
Skeletal muscle of a Kooiker dog featuring a diffuse interstitial and myofibre-directed mononuclear cell infiltration consistent with myositis. Intact myocytes are infiltrated (coring out) by inflammatory cells (arrows). In more severely affected areas, varying stages of the myonecrosis of multiple fibres (frame) with the disruption of the sarcoplasm, invasion, and, later, replacement by macrophages being frequently seen. H&E, a section of skeletal muscle, marked severity, obj. 20×, scale bar = 100 µm.

**Figure 2 animals-14-02519-f002:**
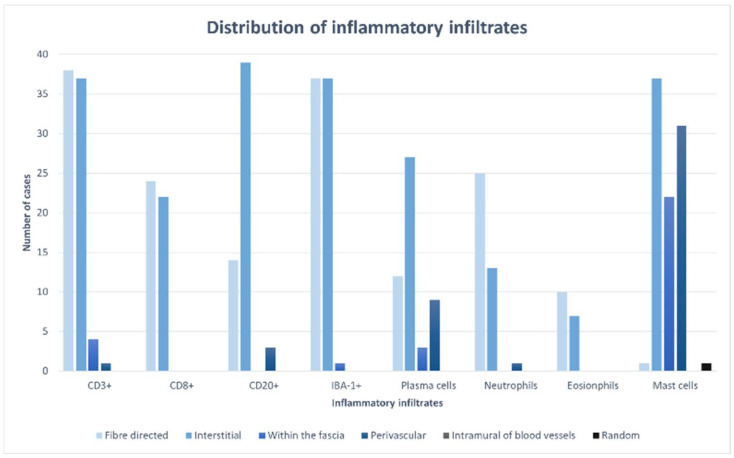
Type and distribution of inflammatory infiltrates.

**Figure 3 animals-14-02519-f003:**
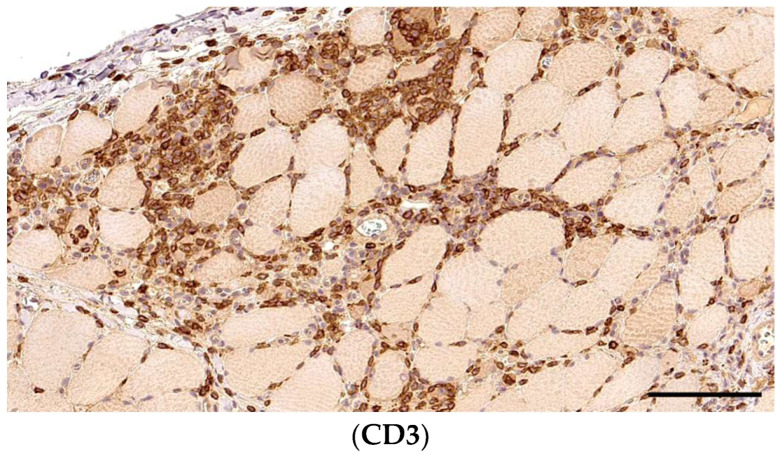
Immunohistochemistry for T-cells with fibre-directed and interstital distributions in the same section of skeletal muscle of Kooiker dogs. In CD3+ T-cells representing the predominant cell type in the Kooiker myositis, they surround and invade non-necrotic myofibres, characteristic of PM. A section of skeletal muscle, obj. 20×, scale bar = 100 µm.

**Figure 4 animals-14-02519-f004:**
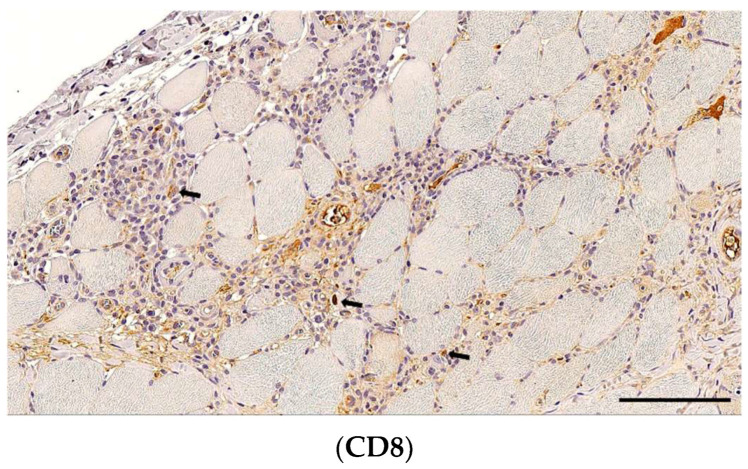
Cytotoxic CD8+ cells scatter alongside the endomysium and in necrotic areas (arrows), albeit in lower number than CD3+ T-cells. A section of skeletal muscle, obj. 20×, scale bar = 100 µm.

**Figure 5 animals-14-02519-f005:**
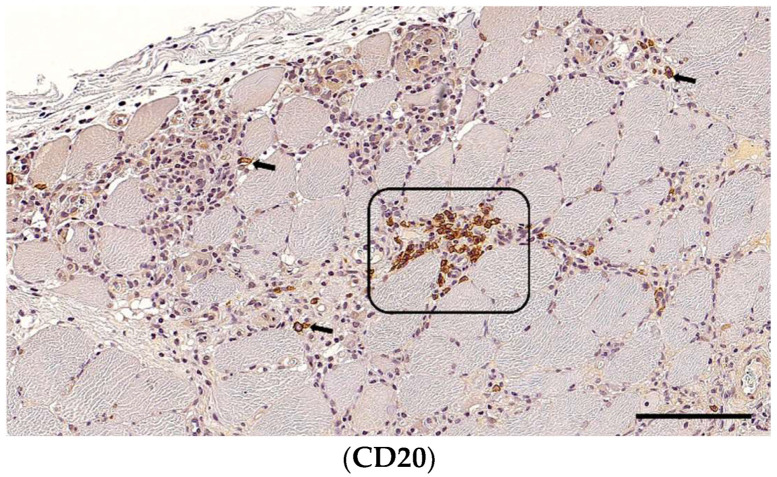
Immunohistochemistry for B-cells and macrophages in the same skeletal muscle of a Kooiker dog. In contrast to previous reports, multifocal B-cell clusters (frame) and scattered B-cells (arrows) were frequently seen. CD20+ B-cells displayed interstitial infiltration and were concentrated in infiltrated and necrotic areas. A section of skeletal muscle, obj. 20×, scale bar = 100 µm.

**Figure 6 animals-14-02519-f006:**
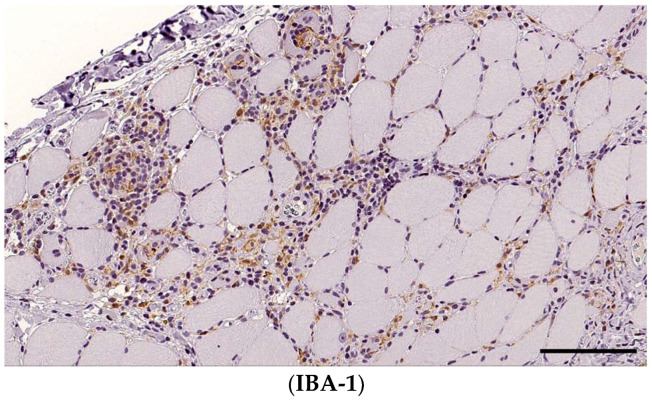
Numerous IBA-1-expressing macrophages encompass non-necrotic myofibres and invade necrotic areas within damaged muscle tissue. A section of skeletal muscle, obj. 20×, scale bar = 100 µm.

**Figure 7 animals-14-02519-f007:**
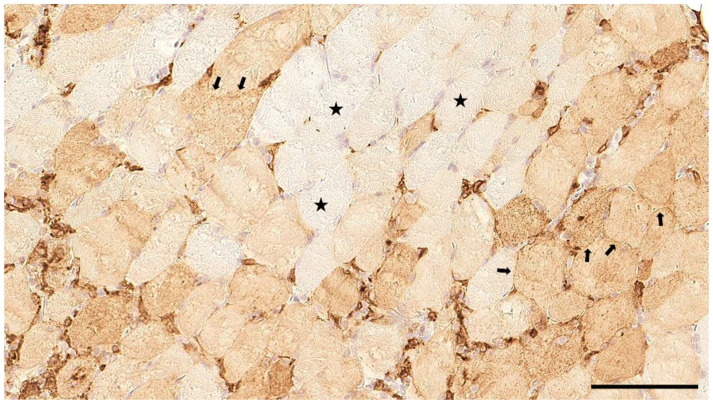
MHC-II immunostaining in the skeletal muscle of a Kooiker dog. Besides APCs, multiple myofibres display a diffuse sarcolemmal MHC-II expression (arrows), whereas other myofibres are unremarkable (asterisks). A section of skeletal muscle, obj. 20×, scale bar = 100 µm.

**Table 1 animals-14-02519-t001:** Exact biopsy sites, as stated, and number of cases.

Biopsy Site	Number of Cases
Muscles of the head	
M. masseter	4
M. temporalis	1
Axial Muscles	
Neck *	1
M. spinalis	1
M. trapezius	2
Ribs *	1
M. iliopsoas	1
Appendicular muscles	
M. supraspinatus	1
M. infraspinatus	1
Shoulder *	3
M. biceps brachii	2
M. triceps	21
Hindlimbs *	2
M. quadriceps femoris	5
M. vastus lateralis	4
M. biceps femoris	14
M. semitendinosus	2
M. gastrocnemius	2
M. tibialis cranialis	4
Nerve *	2

* Localization not further characterized.

## Data Availability

The raw data are available in the Appendix A.

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
