# Peer review of "Pathologic Changes in and Immunophenotyping of Polymyositis in the Dutch Kooiker Dog"

_animals, 2024, doi:10.3390/ani14172519_

Round 1

Reviewer 1 Report

Comments and Suggestions for Authors

The manuscript "Pathologic changes and immunophenotyping of polymyositis in the Dutch Kooiker Dog" by Alf et al. is very well written and it presents scientific merit and contribution in the field of inflammatory miopathies in animals. 

I have few suggestions to improve the presentation of this manuscript:

a) Material and Methods

2.3, lines 137-138 - "puffer" or buffer ? Please check !

2.3 lines 148-149 - Please include the tissues/species used as positive and negative controls

3. Results

3.2, lines 216-222 - This information needs to go to "Material and Methods"

3.2 Fig. 1 - Fig "a" and "b" are very small. I suggest enlarging them to make the data easier to visualize.

Author Response

Reviewer 2:

The manuscript "Pathologic changes and immunophenotyping of polymyositis in the Dutch Kooiker Dog" by Alf et al. is very well written and it presents scientific merit and contribution in the field of inflammatory miopathies in animals. 

I have few suggestions to improve the presentation of this manuscript:

  1. a) Material and Methods

Comment 1: 2.3, lines 137-138 - "puffer" or buffer ? Please check !

Response 1: Thank you for this remark. The German word “puffer” has crept in here and we have overlooked it. We have changed it to the correct “buffer”. (page 3, line 143)

Comment 2: 2.3 lines 148-149 - Please include the tissues/species used as positive and negative controls

Response 2: Thank you for pointing this out. We used canine lymph node tissue as positive and negative control for quality assessment for each antibody. We included this useful information in the manuscript. (page 4, line 154)

  1. Results

Comment 3: 3.2, lines 216-222 - This information needs to go to "Material and Methods"

Response 3: Thank you for this suggestion, we agree with the reviewer. We moved the explanation of our semiquantitative assessment in the appendix.

Comment 4: 3.2 Fig. 1 - Fig "a" and "b" are very small. I suggest enlarging them to make the data easier to visualize.

Response 4: Thank you for this great advice. We decided to remove figure a and focus on figure b.

Reviewer 2 Report

Comments and Suggestions for Authors

Review of Pathologic changes and immunophenotyping of polymyositis in the Dutch Kooiker Dog

In this paper, the authors provide a concise summary of polymyositis (PM) in other canine species and then characterize the histologic lesions of PM in the Dutch Kooiker Dog as well as performing an initial estimate of the primary leukocytes involved in the pathogenesis of this disease.  A great deal of detailed analysis went into this manuscript although the significance of some of the minor distinctions are confusing and several important data points are not provided  (see detailed below) and these issues need to be addressed before publication.  Overall, I am recommending major revision to allow the authors to streamline the pathologic scoring, fill in additional, clinically relevant data points (in particular which anatomic muscle groups are most severely affected), provide a better narrative description of the histologic lesions, and better include support for the MHCII data

Specific issues:

Multiple times the authors state that infiltration is occurring in only “non-necrotic” myocytes (example Line 33) which is incorrect and in fact contradicted by the authors in the caption for Figure 2.  I think the point you are trying to make is that the initial infiltration occurs in intact / viable myocytes with further intensification of inflammation as the cell degenerate (as described in Pathologic Basis of Veterinary Disease, 7th edition page 1033). 

Section 2.2 Histologic evaluation

Line 113: Usually, papers cite the initials of the performing pathologists as well as providing their specific credentials (Diplomate of the American College of Veterinary Pathologists or European College of Veterinary Pathologists)

As an ACVP diplomate who as performed many similar studies, 91 histologic variables are too many to accurately score accurately.  Furthermore, some of these differences seem arbitrary.  For example, polygonal (usually stated as rounded) versus angular atrophy is more a reflection of the numbers of fibers affected rather than a pathologic difference (as described in Pathologic Basis of Veterinary Disease, 7th edition page 1002-1003).  More useful would have been a severity score for the different muscle regions for just atrophy / regeneration / fibrosis etc.

When scoring severity, best practice is to provide a specific rubric for how the scores are given.  For example, minimal = 0-25% of the tissue affected; mild = 26-50% of the tissue affected).  This information needs to be included as you did for the staining grading (Lines 219-222).

Results:

Were there differences in inflammation / degeneration between muscle groups?  This distinction would be extremely helpful for clinicians and pathologists determining which muscle groups to sample.

Lines 228-232: This section is difficult to follow.  A simple narrative stating which cell types were observed in which location in low / moderate / high numbers (rather than the grading) is preferable.

Figure 1 is hard to read, and I am not sure what information the authors are trying to convey.  A simple table with the number of cases of predominately lymphohistiocytic, lymphocytic or neutrophilic inflammation would be more useful.  Or if all the cases were predominately lymphohistiocytic (as seems implied), leave the figure out and just give a narrative description.

Figure 2:  This description could be improved pointing out the rather diffuse interstitial infiltration with viable (not non-necrotic) myocytes with inflammatory cell infiltrates and areas of more intense inflammation associated with necrotic fibers with focal effacement. (“cored out” is a non-standard description)

Figure 3:  There is a lot of background staining making interpretation a little difficult, but I agree with the overall assessment of cell type.  However, it does appear in 3a that some of the inflammation is located within necrotic or at least degenerating myocytes.  A more accurate description (with arrows) would indicate infiltration of adjacent viable myocytes next to the region of more dense inflammation

If possible, I would suggest a single panel of four images with the four different stains from the same muscle region, but I realize this ideal may not be practical.  Alternatively, combine all four images into one panel for direct comparison which would show that the initial step is invasion by CD3+ cells.

Figure 6.  There is a great deal of background staining in this image, and I am not convinced the staining is not largely within antigen presenting cells (APC).  Only the left most arrow appears to be pointing to cytoplasmic staining but even that focus is not definitive focal cytoplasmic staining of a myocyte and not a portion of an APC.  Perhaps pairing with a higher magnification with better blocking to illustrate cytoplasmic staining.  Otherwise, this data should be removed.

 Discussion:

The discussion covers most of the important points, but I would suggest including the “unique” features in the first paragraph as part of the general summary.

Author Response

Reviewer 3:

Review of Pathologic changes and immunophenotyping of polymyositis in the Dutch Kooiker Dog

In this paper, the authors provide a concise summary of polymyositis (PM) in other canine species and then characterize the histologic lesions of PM in the Dutch Kooiker Dog as well as performing an initial estimate of the primary leukocytes involved in the pathogenesis of this disease.  A great deal of detailed analysis went into this manuscript although the significance of some of the minor distinctions are confusing and several important data points are not provided  (see detailed below) and these issues need to be addressed before publication.  Overall, I am recommending major revision to allow the authors to streamline the pathologic scoring, fill in additional, clinically relevant data points (in particular which anatomic muscle groups are most severely affected), provide a better narrative description of the histologic lesions, and better include support for the MHCII data

Specific issues:

Comment 1: Multiple times the authors state that infiltration is occurring in only “non-necrotic” myocytes (example Line 33) which is incorrect and in fact contradicted by the authors in the caption for Figure 2.  I think the point you are trying to make is that the initial infiltration occurs in intact / viable myocytes with further intensification of inflammation as the cell degenerate (as described in Pathologic Basis of Veterinary Disease, 7th edition page 1033). 

Response 1: Thank you for addressing this point and making us aware of possible misunderstandings originating from this phrasing. We changed all passages accordingly. (page1, lines 33-34; page 2, lines 82 - 83)

Comment 2: Section 2.2 Histologic evaluation

Line 113: Usually, papers cite the initials of the performing pathologists as well as providing their specific credentials (Diplomate of the American College of Veterinary Pathologists or European College of Veterinary Pathologists)

Response 2: Thank you for this suggestion, we added the detailed information about our diagnosticians. (page 3, lines 114-155 and lines 124-125)

Comment 3: As an ACVP diplomate who as performed many similar studies, 91 histologic variables are too many to accurately score accurately.  Furthermore, some of these differences seem arbitrary.  For example, polygonal (usually stated as rounded) versus angular atrophy is more a reflection of the numbers of fibers affected rather than a pathologic difference (as described in Pathologic Basis of Veterinary Disease, 7th edition page 1002-1003).  More useful would have been a severity score for the different muscle regions for just atrophy / regeneration / fibrosis etc.

Response 3: We appreciate the reviewer´s insight, suggestions and the references provided. Indeed 91 histologic variables represents many parameters, but the description of polymyositis in this cohort is a spinoff of a broader analysis characterizing canine inflammatory myopathies that is in its writing phase. We enhanced the diagnostic algorithm for neuromuscular disorders, for which the subtleties of these parameters are essential to come to an aetiologic diagnosis. Polygonal atrophic fibres are an unspecific finding shared by many acquired and congenital muscle disorders, while angular atrophy is suggestive of disuse or denervation. As the reviewer knows, semiquantitative scores have intrinsic limitations, but we believe that in a descriptive paper as this correctly reporting the qualitative changes is helpful for the diagnosticians that do not have a strong background in neuromuscular disorders.

Comment 4: When scoring severity, best practice is to provide a specific rubric for how the scores are given.  For example, minimal = 0-25% of the tissue affected; mild = 26-50% of the tissue affected).  This information needs to be included as you did for the staining grading (Lines 219-222).

Response 4: Fully agree with the reviewer. We implemented the specification for the extension of these changes in the appendix together with the explanations for the other scores. Thank you for highlighting this missing information. 

Comment 5: Results:

Were there differences in inflammation / degeneration between muscle groups?  This distinction would be extremely helpful for clinicians and pathologists determining which muscle groups to sample.

Response 5: Thank you for this suggestion. We had also discussed this matter when analysing our data, but unfortunately, we had to realise that retrospectively, we could not retrieve this data. As this is a retrospective study and some of our cases date back to the 90s, we were unable to clearly trace the topography of all muscle collected or to obtain this information afterwards. Either the exact location of the biopsy was not indicated, and/or the individual samples were not clearly labelled. However, we also believe that this information would be of great added value.

Comment 6: Lines 228-232: This section is difficult to follow.  A simple narrative stating which cell types were observed in which location in low / moderate / high numbers (rather than the grading) is preferable.

Response 6: Thank you for insight, we modified the text accordingly and hopefully this phrasing makes the listing easier to read. (page 6, lines 223-235)

Comment 7: Figure 1 is hard to read, and I am not sure what information the authors are trying to convey.  A simple table with the number of cases of predominately lymphohistiocytic, lymphocytic or neutrophilic inflammation would be more useful.  Or if all the cases were predominately lymphohistiocytic (as seems implied), leave the figure out and just give a narrative description.

Response 7:

We appreciate this feedback, thank you. We have decided to remove Figure 1a and instead focus on Figure 1b to provide a concise overview and summary of the inflammatory infiltrates observed. (page 6, lines 239 – 241)

Comment 8: Figure 2:  This description could be improved pointing out the rather diffuse interstitial infiltration with viable (not non-necrotic) myocytes with inflammatory cell infiltrates and areas of more intense inflammation associated with necrotic fibers with focal effacement. (“cored out” is a non-standard description)

Response 8:

As recommended by the reviewer, we modified the legend accordingly. Also, we cross checked our original reference for cored-out fibers from the Jubb Kennedy and Palmer’s, Pathology of Domestic Animals 6th Edition, Volume 1, Chapter 3, page 226, Figure 3-83. The reported description is coring out, not cored-out hence we modified that throughout the manuscript. (page 5, line 203; page 7, line 276; page 9, line 312; Appendix A; Appendix B)

Comment 9: 

Figure 3:  There is a lot of background staining making interpretation a little difficult, but I agree with the overall assessment of cell type.  However, it does appear in 3a that some of the inflammation is located within necrotic or at least degenerating myocytes.  A more accurate description (with arrows) would indicate infiltration of adjacent viable myocytes next to the region of more dense inflammation

Comment 10:  

If possible, I would suggest a single panel of four images with the four different stains from the same muscle region, but I realize this ideal may not be practical.  Alternatively, combine all four images into one panel for direct comparison which would show that the initial step is invasion by CD3+ cells.

Comment 11: 

Figure 6.  There is a great deal of background staining in this image, and I am not convinced the staining is not largely within antigen presenting cells (APC).  Only the left most arrow appears to be pointing to cytoplasmic staining but even that focus is not definitive focal cytoplasmic staining of a myocyte and not a portion of an APC.  Perhaps pairing with a higher magnification with better blocking to illustrate cytoplasmic staining.  Otherwise, this data should be removed.

Response 9/10/11:

We greatly appreciate the reviewers’ constructive comments regarding our picture selection. Thank you for your valuable feedback. To enhance the quality of the manuscript, we have decided to follow the reviewers' suggestion to retake our pictures from the same muscle region with four different stains for direct comparison. Additionally, we have replaced the MHC-II figure with an improved version.

Comment 12:  Discussion:

The discussion covers most of the important points, but I would suggest including the “unique” features in the first paragraph as part of the general summary.

Response 12:

Thank you for this suggestion. We added a summary of our main findings on page 9, lines 314 – 319 at the beginning of our discussion.